# Peripheral Modulators of the Central Fatigue Development and Their Relationship with Athletic Performance in Jumper Horses

**DOI:** 10.3390/ani11030743

**Published:** 2021-03-08

**Authors:** Francesca Arfuso, Claudia Giannetto, Elisabetta Giudice, Francesco Fazio, Michele Panzera, Giuseppe Piccione

**Affiliations:** Department of Veterinary Sciences, University of Messina, Polo Universitario dell’Annunziata, 98168 Messina, Italy; farfuso@unime.it (F.A.); egiudice@unime.it (E.G.); ffazio@unime.it (F.F.); mpanzera@unime.it (M.P.); gpiccione@unime.it (G.P.)

**Keywords:** jumping exercise, horses, amino acids, athletic performance, serum dopamine, serum prolactin

## Abstract

**Simple Summary:**

Physical exercise induces various stress responses leading to a disturbance of homeostasis and a number of regulatory systems are called upon to return the body to a new level of equilibrium. The aim of this study was to evaluate the peripheral modulators of serotoninergic function and neurohumoral factors’ changes in athletic horses during an official jumping competition, and to evaluate their relationship with the physical performance of competing horses. The findings obtained in the current survey showed that jumping exercise influenced the levels of tryptophan, leucine, valine, non-esterified fatty acids (NEFAs), dopamine, and prolactin and that these changes are related to the physical performance of competing horses. These findings suggest that the serotoninergic system may be involved in fatigue during jumping exercise.

**Abstract:**

The current study aimed to investigate whether peripheral modulators of serotoninergic function and neurohumoral factors’ changes in athletic horses during an official jumping competition, and to evaluate their relationship with the physical performance of competing horses. From 7 Italian Saddle mares (6–9 years; mean body weight 440 ± 15 kg), performing the same standardized warm-up and jumping course during an official class, heart rate (HR) was monitored throughout the competition. Rectal temperature (RT) measurement, blood lactate and glucose concentration, serum tryptophan, leucine, valine, the tryptophan/branched-chain amino-acids ratio (Try/BCAAs), dopamine, prolactin, and non-esterified fatty acids (NEFAs) were assessed before the exercise event (T0), at the end of the competition stage (5 min ± 10 s following the cessation of the exercise, T_POST5_), and 30 min after the end of competition (T_POST30_). Highest HR values were recorded during the course and at the outbound (*p* < 0.0001); blood lactate concentration and RT increased after exercise with respect to the rest condition (*p* < 0.0001). Lower leucine and valine levels (*p* < 0.01), and higher tryptophan, Try/BCAAs ratio, and NEFAs values were found at T_POST5_ and T_POST30_ with respect to T0 (*p* < 0.0001). A higher prolactin concentration was found at T_POST5_ and T_POST30_ compared to T0 (*p* < 0.0001), whereas dopamine showed decreased values after exercise compared to rest (*p* < 0.0001). Statistically significant correlations among the peripheral indices of serotoninergic function, neurohumoral factors, and athletic performance parameters were found throughout the monitoring period. The findings provide indirect evidence that the serotoninergic system may be involved in fatigue during jumper exercise under a stressful situation, such as competition, in which, in addition to physical effort, athletic horses exhibit more passive behavior.

## 1. Introduction

Animals competing in equestrian disciplines, particularly in show jumping, are required to have high technical skills. Among the parameters considered for the evaluation of athletic performance in horses, heart rate and blood lactate concentration are the most studied as these parameters are good indices of the fitness level and of the workload effort. Cardiovascular, respiratory, and metabolic responses and musculoskeletal adaptations to exercise have been well studied in equine species, allowing specific exercise testing for racehorses [1]. Nowadays, new markers of performance are used in racehorses, including the evaluation of peripheral blood mononuclear cell proliferation and activity or cytokines’ mRNA expression. Changes in immune cell proliferation, lymphocyte populations, and monocyte functionality have been described in trained and untrained racehorses after exercise, confirming the creation of an anti-inflammatory environment in well-trained horses [2]. It has been suggested that long-distance endurance rides involve strenuous effort, which induces numerous changes in the horse’s body, including the exercise-induced acute-phase response [2,3]. Moreover, regular physical activity results in a decrease of proinflammatory states [3]. Although studies have dealt with exercise testing for event [4], dressage [5], and jumper horses [6], a paucity of information is available on the physical performance and fatigue signals arising in athletic horses during official competition.

Despite its current popularity, the cause(s) of fatigue and its underlying mechanism(s) have only recently begun to be explored in the athlete horse. Fatigue is an important factor affecting exercise and sporting performances. It is defined physiologically as the inability to maintain power output [7,8], and the organism uses it as a defense mechanism to avoid irreversible damage due to excessive exertion. Fatigue during exercise has traditionally been considered to be the result of events localized within skeletal muscle [9]. However, fatigue cannot always be ascribed to events residing within skeletal muscle; indeed, it is a complex multifactorial element with peripheral and central components. Central fatigue develops in the central nervous system and involves brain serotonin levels [10]. Although it seems naive to assume that the activity of serotoninergic neurons should be determined by substrate availability alone, the central fatigue hypothesis proposes that this situation may arise during periods of stress, such as those occurring during exercise and/or competition. The serotonergic system is associated with numerous brain functions that can positively or negatively affect endurance [11] Accordingly, the synthesis and metabolism of serotonin in the brain increases in response to exercise [12]. Furthermore, the rise of brain serotonin concentration is associated with markers of central fatigue, such as decreased motivation, lethargy, tiredness, and loss of motor coordination [11]. An increase of serotonin synthesis in the brain is correlated with high levels of blood-borne tryptophan, the amino acid precursor to serotonin. The rate-limiting step in the synthesis of serotonin is the transport of tryptophan across the blood–brain barrier into the brain [13].

Besides tryptophan, other compounds in the bloodstream, such as branched-chain amino acids (BCAAs), lipids, hormones, and neurotransmitters, can orchestrate the fine mechanism of the onset of fatigue. During and after exercise, BCAAs intervene in muscle protein synthesis by stimulating mRNA translation [14] and prevent muscle proteolysis by inhibiting mechanisms that involve the mammalian target of rapamycin [15,16]. The increased uptake of BCAAs and their oxidation by skeletal muscle will reduce the concentration, available for competition, with tryptophan for transport across the blood–brain barrier, resulting in an increase in the uptake of tryptophan to the brain, and therefore the synthesis and release of serotonin by the brain [17]. Studies carried out on human species [18,19,20,21] have shown that, depending on the intensity and duration, exercise stimulates the release of the hormone prolactin as well as dopamine synthesis, resulting in behavioral and physiological changes. Indeed, emotional or physical stress can temporarily increase prolactin levels. Prolactin also interacts with the dopaminergic system and has been linked to anxiety in several species. The influence of neurotransmitters on fatigue has also been proposed, and dopamine seems to be linked to the ‘central’ component of fatigue for its well-known role on motivation and motor behavior and it is therefore thought to have an enhancing effect on performance [22,23].

Although show jumping is part of the Olympic equestrian disciplines, and despite the need to monitor fitness and workload, there is a paucity of information about field exercise testing in jumper horses during official competition [6,24] and none of these tests have assessed the indices of serotoninergic function and their relationship with exercise performance.

In view of these considerations, the current study aimed to investigate the effects of jumping exercise on some peripheral modulators of serotoninergic function (e.g., tryptophan, tryptophan/BCAAs ratio, NEFAs) and neurohumoral factors (e.g., dopamine, prolactin) as well as their relationship with physical performance in high-level jumper horses performing the same standardized warm-up and jumping course during an official competition.

## 2. Materials and Methods

### 2.1. Animals and Experimental Design

This study was carried out on 7 Italian Saddle mares (6–9 years; mean body weight 448 ± 15 kg), after the informed consent of the owners. All horses were managed equally at the same horse training center in Sicily, Italy (latitude 38°10′35′′ N; longitude 13°18′14′′ E), housed in individual boxes (3.5 × 3.5 m), under natural photoperiod and environmental conditions (mean temperature of 26 ± 5 °C and mean relative humidity of 66 ± 4%). All horses were trained and ridden by the same trainer and rider, respectively. The diet composition for the horses was formulated according to their training requirements. In particular, animals were fed twice a day (7.00 a.m.; 5.00 p.m.), with a total food amount of about 2.5% dry-mater of horse body weight (forage: concentrate ratio 70:30), and water was available ad libitum. All animals were clinically healthy (based on a thorough clinical examination) and free from internal and external parasites. Horses enrolled in the current study had the same level of training, similar fitness, and the same experience of jumping competition.

The horses took part in an outdoor jumping competition. The session was preceded by a warm-up on the flat consisting of walk 1 (1 min), trot (3 min), walk 2 (1 min), canter (2 min), and walk 3 (1 min). The next stage of standardized warm-up included 4 vertical and 4 oxer jumps of increasing height (height: from 100 to 140 cm). The trainer timed each warm-up stage and used the same sequence of jumps for all subjects (alternating verticals and oxers with the same eight). After the warm-up, horses competed in the same jumping course with the following technical specifications: total length, 500 m; obstacle height, 140 cm; and total efforts, 15 (9 verticals, 6 oxers, 1 double combination, 1 triple combination). The competition stage was made up of four phases, including: inbound (waiting time inside the arena before competing), course (time of the jumping phase), outbound (time lapse between the end of the course and the exit from the arena), and end (time lapse between the exit and the arrival to the stable).

### 2.2. Rectal Temperature (RT) and Heart Rate (HR) Measurement 

The RT, taken as being representative of the body temperature, was measured by means of a digital thermometer (HI92704, Hanna Instruments, Leighton Buzzard, Bedfordshire, UK), inserted 15 cm in the rectum before the exercise event (T0), at the end of competition stage (5 min ± 10 s following the cessation of the exercise, T_POST5_), and 30 min after the end of the competition (T_POST30_).

To evaluate the workload during the competition, each horse was equipped with equine HR monitors (Polar Horse Trainer, S 610, Polar Electro Europe BV, Fleurier Branch, Avenue Daniel-Jeanrichard, Switzerland) to record HR during each step of the warm-up and competition. Two electrodes were placed against the horse’s wet coat: the positive electrode was first placed under the saddlepad, the negative electrode was then fixed to the saddle girth on the left side of the thorax, and finally the electrodes were connected to a transmitter (T51H, Polar Electro Oy, Kempele, Finland), fixed to a breast strap, that transmitted data to a watch-type data logger (Polar S-610I, Polar Electro Oy, Kempele, Finland), placed near the electrodes. Recorded data were then downloaded on a personal computer, by using the Polar Equine 4.0 software (Kempele, Finland), to be analyzed. In particular, HR was logged every 5 s during the whole exercise. The values of HR measured before the exercise event at rest (T0); during each step of warm-up, including walk, trot, canter, and jumps 1–8; and competition, including inbound, course, outbound, and end of the competition stage (5 min ± 10 s) following the cessation of the exercise, T_POST5_); and 30 min after the end of the competition (T_POST30_).

### 2.3. Blood Sampling Procedures and Laboratory Analysis

Blood collection was performed by the same operator before the exercise event (T0), at the end of the competition stage (5 min ± 10 s following the cessation of the exercise, T_POST5_), and 30 min after the end of the competition (T_POST30_). The samples were collected by jugular venipuncture into EDTA vacuum test tubes (Terumo Corporation, Tokyo, Japan) to assess blood lactate and glucose concentration immediately after collection using a small handheld meter (Accutrend Plus, Roche Diagnostics, Deutschland, Germany). Moreover, from each subject, blood samples were collected into three vacuum tubes with clot activator (Terumo Corporation, Tokyo, Japan), which were placed in refrigerated bags and transported to the laboratory for analysis. One tube with clot activator was centrifuged at 1300g for 10 min and the individual serum samples obtained were deproteinized with 5-sulfosalicylic acid and leucine, valine, and tryptophan concentrations were assessed by the high-performance liquid chromatography method. The ratio between tryptophan and BCAAs levels (Try/BCAAs) was then calculated for each sample. On serum samples, the concentration of serum total proteins (biuret method) and NEFAs was determined by means of commercially available kits (total proteins, Byosistems, Reagents and Instruments, Barcelona, Spain; NEFAs, Randox, Crumlin, UK) by means of an automated analyzer ultraviolet-visible spectrophotometer (model Slim SEAC, Florence, Italy).

The second tube with cloth activator was centrifuged at 1000× *g* for 20 min for the assessment of serum dopamine concentration, whereas for the measurement of serum prolactin concentration, the third tube with cloth activator was centrifuged at 1000× *g* for 15 min. Serum dopamine concentration was assessed using a commercially available kit by the quantitative Sandwich ELISA method with a sensitivity of 1.0 pg/mL, a detection range of 6.25–200 pg/mL, and both an intra-assay and inter-assay coefficient of variability of less than 15%. Serum prolactin concentration was assessed using a commercially available kit by the quantitative Sandwich ELISA method with a sensitivity of 3.1 ng/mL, a detection range of 6.25–400 ng/mL, and an intra-assay coefficient of variability of less than 8%, and inter-assay coefficient of variability of less than 12%.

### 2.4. Statistical Analysis

Before statistical analysis, all data were tested for normality of distribution using the Kolmogorov–Smirnov test. All data were normally distributed (*p* > 0.05) and parametric statistical analysis was performed. One-way repeated measures analysis of variance (ANOVA) was applied to determine statistically significant differences in the values of RT, HR, blood lactate and glucose concentration, serum tryptophan, leucine, valine, Try/BCAAs, total proteins, NEFAs, dopamine, and prolactin obtained from each horse during the monitoring period. The Bonferroni multiple comparison test was applied for post hoc comparison.

The Pearson’s correlation test was performed to assess whether the peripheral modulators of serotorinergic function (e.g., tryptophan, leucine, valine, Try/BCCAs, NEFAs) correlated with themselves, and with neurohumoral factors (e.g., dopamine and prolactin), as well as with indices of athletic performance in jumper horses throughout the monitoring period.

*p*-Values < 0.05 were considered statistically significant. Data were analyzed using statistical software Prism v. 4.00 (Graphpad Software Ldt, San Diego, CA, USA, 2003).

## 3. Results

All the results are expressed as mean values ± standard deviation (±SD).

As shown in Figure 1, higher HR values were found at T_POST5_ compared to T0 and T_POST30_ at each step of the competition (*p* < 0.0001). Moreover, the highest HR values were recorded during the course and at the outbound. The statistical analysis of data of HR recorded during the jumping stage of the warm-up showed a significant increase of HR values when shifting from 100- to 125-cm jumps (*p* < 0.001). As shown in Figure 2, blood lactate concentration showed higher values at T_POST5_ with respect to T0 and T_POST30_ (*p* < 0.001), and RT also increased after exercise (T_POST5_ and T_POST30_) with respect to the rest condition (*p* < 0.0001), whereas the total protein and glucose concentration was unchanged after exercise compared to the rest condition (*p* > 0.05). According to the findings gathered from the application of one-way ANOVA and as shown in Figure 3, leucine and valine showed lower levels (*p* < 0.01) after exercise (T_POST5_ and T_POST30_) compared to baseline values measured at rest (T0), whereas higher tryptophan, Try/BCAAs ratio, and NEFAs values were found at T_POST5_ and T_POST30_ with respect to T0 (*p* < 0.0001). A higher prolactin concentration was found at T_POST5_ and T_POST30_ compared to T0 (*p* < 0.0001), whereas dopamine showed lower values after exercise compared to rest (*p* < 0.0001).

Statistically significant correlations among the peripheral indices of serotoninergic function, neurohumoral factors, and athletic performance parameters were found throughout the monitoring period (Table 1).

## 4. Discussion

Physical exercise, depending on the type and intensity, induces various stress responses leading to a disturbance of homeostasis and a number of regulatory systems are called upon to return the body to a new level of equilibrium [5]. The nervous and endocrine systems work in concert to initiate and control movement, and all involved physiological processes. According to the findings obtained in the current study, the main indices of horses’ fitness showed dynamic changes throughout the monitoring period. Specifically, in agreement with previous studies investigating horse performance during official jumping competition [24,25], the results of the current study confirmed the significant contribution of anaerobic metabolism during jumping exercise [1], showing an increase of the blood lactate concentration 5 min after completing the exercise followed by a decline 30 min after the end of competition. As a matter of fact, although within a few minutes lactate release started to decrease because the removal from the circulation exceeds the release rate from the muscles [26], it is known that the clearance of lactic acid from the muscles and bloodstream takes 30 min after exercise in well-trained horses [27]. The HR of each subject was influenced by each step of both the warm-up and jumping course of the competition, showing the highest values during and immediately after the jumping course. Noteworthy, the high HR values registered during the inbound to the show jumping arena highlighted that the official competition itself represented a contributing factor to HR increase as mental stress [28] and emotionality [29]. As a matter of fact, the stress response after training sessions and competitions in horses is different and HR may indicate the animal’s internal physiological state; indeed, the rise in HR is not only caused by an increase in physical activity, but also reflects heightened emotional reactivity [29]. The trend of HR registered in jumper horses of the current survey agrees with the observations of a previous study carried out on Belgian Saddle breeds competing in the Belgian Championship [24] in which it was suggested that horses experienced a need for acceleration near the finish of the competition, causing HR increase due to the onset of fatigue. Furthermore, the results gathered in the present study highlighted that the obstacle height caused an increase of HR values in competing horses. The rise in HR could be due to sympathetic nervous activity, which is known to increase with increasing exercise intensity, leading to a consistent rise of catecholamine levels. It is well established that the increase of the catecholamine concentration enhances the force of cardiac contraction and thus the cardiac output and the increase of HR. The results obtained in this study showed a significant increase of RT after physical exercise and this was not surprising. Indeed, during exercise, horses’ muscles work and a conversion of stored chemical energy into mechanical energy and thermal energy occurs [30,31]. The heat produced by muscle during exercise raises the core temperature. Therefore, heat dissipation mechanisms should be activated, and nowadays, the most developed techniques, such as infrared thermography, which measures the muscle temperature changes and correlates with lactate concentration, are often used in race training [32]. The primary physiologic mechanism leading to heat dissipation is represented by the increase in cardiac output that results in heat transfer from the core to the cutaneous blood circulation by eliciting reflex neurogenic vasodilatation in the skin, resulting in increased skin temperature [30].

Body temperature is controlled by the opposing effects of adrenergic and serotoninergic neuronal activity in the hypothalamus. A rise in body temperature may be due to augmented serotoninergic activity combined with inhibition of adrenergic activity, or purely due to increased serotoninergic activity [33]. Noteworthy, the results found herein seem to suggest that the potential peripheral modulators of serotoninergic function measured in athletic horses during a jumping competition tended to respond as predicted in the “central fatigue hypothesis” by Newsholme et al. [10]. The levels of leucine, valine, and dopamine decreased after exercise in competing horses, whereas tryptophan, Try/BCAAs ratio, prolactin, and NEFAs showed increased concentrations following jumping competition exercise. It could be hypothesized that the BCAAs are used by muscles, mainly active skeletal muscle, during short-term intense exercise as a source of energy for muscle contraction [8]. The increase of tryptophan found after exercise could be related to the rise of the lipid content in the blood, which is likely to occur during exercise, as confirmed by the positive correlation found between the values of this amino acid and NEFAs throughout the monitoring period. During physical exercise, free fatty acids are mobilized from adipose tissue and are transported via the blood to the muscle, where they serve as fuel for muscle contraction [34]. Both free fatty acids and tryptophan compete for binding to albumin and thus, the increase of the NEFAs concentration during exercise, resulting from an elevated adrenergic tone that stimulates subcutaneous lipolysis [35], could limit the binding of tryptophan and albumin, increasing the concentration of serum free tryptophan. The uptake of BCAAs by the active muscle as a source of energy yields an increase of the tryptophan/BCAA ratio after exercise [36]. The decrease in the BCAAs blood serum levels and the increase of the tryptophan level after jumper exercise could increase the rate of passage of tryptophan across the blood–brain barrier because BCAAs and tryptophan compete for the same transporter [37,38,39]. Thus, enhanced entry of tryptophan into the brain would lead to increased serotonin levels, which ultimately leads to anticipated central fatigue [36]. The changes in the values of peripheral modulators of serotoninergic function found in competing horses after exercise showed a relationship with the considered indices of exercise performance, as emphasized by the positive correlation found between the serum levels of tryptophan, Try/BCCAs, and NEFAs, and the values of blood lactate, HR, and RT. On the other hand, the amino acids leucine and valine showed a negative correlation with the values of blood lactate, HR, and RT measured in competing horses affect exercise. Particularly noteworthy, the peripheral modulators of serotoninergic function were negatively correlated with dopamine and positively correlated with prolactin, suggesting a possible linkage among the central fatigue and the neurohumoral factors. Dopamine, acting on specific brain areas, is related to the motivation for locomotion [23]. Exercise may elevate the global dopamine level in the brain without affecting cerebral release via the jugular venous blood, because polar catecholamines do not readily penetrate the blood–brain barrier [40,41], and maybe only dopamine released into the hypophysial portal blood will appear in the jugular blood [42]. However, in healthy subjects, the brain demonstrates noradrenaline spillover into both the major and the minor jugular vein [43], and, although the passage of monoamines from the bloodstream to the brain is restricted by the blood–brain barrier, the existence of a barrier to movement in the opposite direction is less certain [44]. It has been suggested that the main source of dopamine in circulating blood could be the activity of the adrenal medulla [45]. Although what occurs in the central nervous system is known, little information is available from the literature about the effect of exercise on dopamine levels at the peripheral level. The decreased dopamine values found in the present study after the competition conclusion could be due to the transition of circulating dopamine in the spleen, likely to occur during a stressful condition [46]. In the spleen, the noradrenergic sympathetic fibers can take up circulating dopamine, which could then be released on sympathetic activation to be taken up in turn by leukocytes through active transport [47]. The finding of the current study showed an increase of prolactin levels after jumping competition that, hypothetically, could be due to an immediate antagonism of hypothalamic dopamine input to the pituitary, or to some other stimulation factors related to the hypothalamic–hypophysial portal system [18,48]. Thus, it could be assumed that the higher prolactin values recorded after competition are related to the superimposition of stimulatory input of prolactin-releasing factors, such as thyrotrophin-releasing hormone, arginine vasopressin, vasoactive intestinal peptide, and serotonin. Particularly noteworthy, the values of prolactin were positively correlated with the fitness and stress indices, including HR, blood lactate, and RT, recorded from competing horses; however, whether this hormone is involved in the increased perception of effort and the earlier onset of fatigue during exercise remains to be determined. It has been proposed that exercise-induced hyperprolactinemia was related to changes in the peripheral modulators of serotonergic function in human species [49]. However, studies on the possible relationship of prolactin secretion and peripheral modulators of serotoninergic function during exercise showed conflicting results. Fischer et al. [50] found an increase in prolactin concentration during exercise in proportion to the rise in the plasma free tryptophan concentration. On the other hand, prolactin secretion during exercise has subsequently been reported to be unrelated to the peripheral modulators of serotoninergic function [51].

## 5. Conclusions

To the best of the authors’ knowledge, this is the first study investigating the effect of jumping exercise on peripheral modulators and indices of serotoninergic function and their relationship to exercise performance in horses competing in an official jumping class. The findings provide indirect evidence that the serotoninergic system may be involved in fatigue during jumping exercise under a stressful situation, such as competition, in which, in addition to physical effort, athletic horses experience emotional reactivity and mental stress. Although a relationship among peripheral modulators, neurohormones, and indices of fitness seems to exist during jumping exercise in horses, further studies are demanded to determine whether this linkage is responsible for the increased perception of effort and the earlier onset of fatigue during both training and competition in athletic horses. This would contribute to an improvement of the knowledge on horse’s response to jumping effort, and allow a better understanding of how a horse is coping with its training exercise or passive behavior during a competition, which will in turn contribute to improving animal welfare and athletic performance.

## Figures and Tables

**Figure 1 animals-11-00743-f001:**
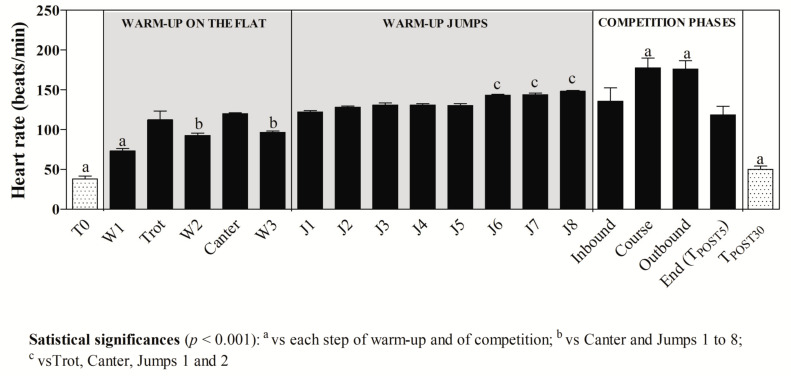
Heart rate values recorded from each horse during the warm-up and competition. The warm-up on the flat included a walk (W1–W3), trot, and canter section followed by 8 warm-up jumps (J1–J8). The competition in the show jumping arena included inbound (waiting time inside the arena before competing), course (time of the jumping phase), outbound (time lapse between the end of the course and the exit from the arena), and end (time lapse between the exit and the arrival to the stable).

**Figure 2 animals-11-00743-f002:**
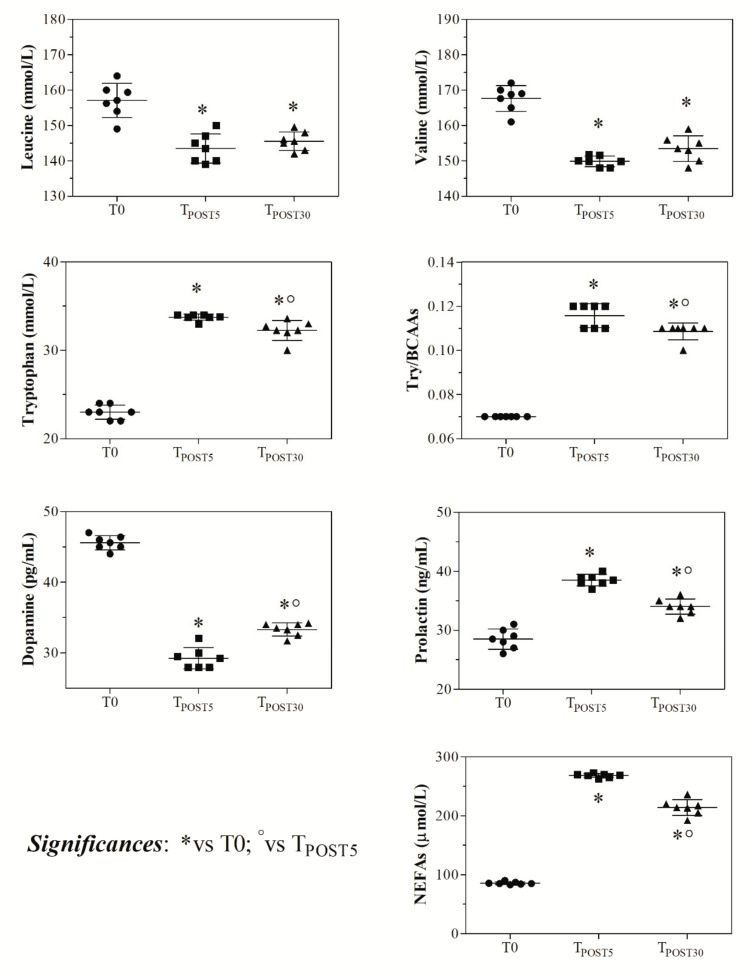
Blood lactate and glucose concentration, serum total protein, and rectal temperature values measured in each competing horse before the exercise event (T0), at the end of the competition stage (5 min ± 10 s) following the cessation of the exercise, T_POST5_), and 30 min after the end of the competition (T_POST30_).

**Figure 3 animals-11-00743-f003:**
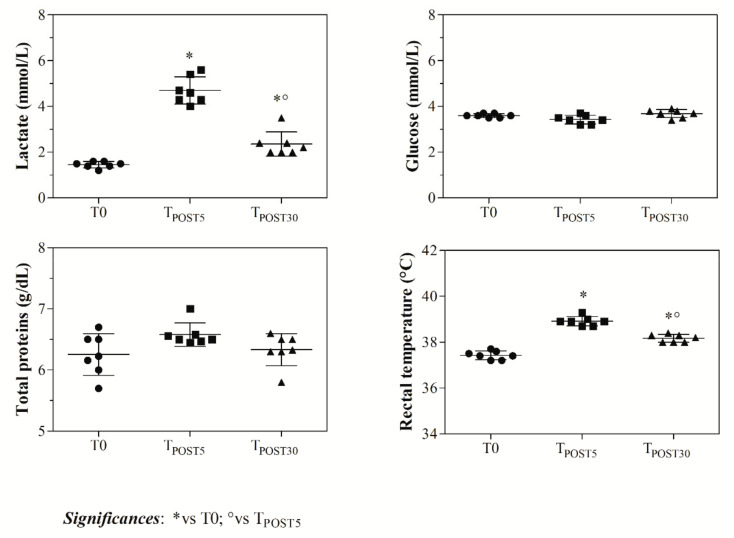
Serum amino acids (e.g., tryptophan, leucine, valine), Try/BCAAs ratio, NEFAs, dopamine, and prolactin values measured in each competing horse before the exercise event (T0), at the end of the competition stage (5 min ± 10 s) following the cessation of the exercise, T_POST5_), and 30 min after the end of the competition (T_POST30_).

**Table 1 animals-11-00743-t001:** Coefficients of correlation among peripheral modulators of serotoninergic function (e.g., tryptophan; leucine; valine; Try/BCCAs ratio; NEFAs), neurohumoral factors (e.g., dopamine and prolactin), and indices of athletic performance (e.g., heart rate, HR; rectal temperature, RT; blood lactate concentration; blood glucose concentration) in jumper horses throughout the monitoring period. Values and age calculated for males and females of foals and for males and females of horses during the monitoring period. *p*-Values < 0.05 were considered statistically significant.

Parameters-	Try/BCAAs	Prolactin (ng/mL)	Dopamine (pg/mL)	NEFAs (μmol/L)	HR (beats/min)	RT (°C)	Lactate (mmol/L)	Glucose (mmol/L)
Tryptophan (mmol/L)	*r* = 0.99 *p* < 0.0001	*r* = −0.26 *p* = 0.29	*r* = −0.97 *p* < 0.0001	*r* = 0.75 *p* < 0.0001	*r* = 0.69 *p* = 0.002	*r* = 0.88 *p* < 0.0001	*r* = 0.75 *p* < 0.0001	*r* = −0.26 *p* = 0.29
Leucine (mmol/L)	*r* = −0.84 *p* < 0.0001	*r* = −0.26 *p* = 0.29	*r* = 0.79 *p* = 0.0001	*r* = 0.75 *p* < 0.0001	*r* = −0.57 *p* = 0.01	*r* = −0.66 *p* = 0.002	*r* = −0.69 *p* = 0.001	*r* = 0.5 *p* = 0.84
Valine (mmol/L)	*r* = −0.90 *p* < 0.0001	*r* = −0.26 *p* = 0.29	*r* = 0.89 *p* < 0.0001	*r* = 0.75 *p* < 0.0001	*r* = −0.69 *p* = 0.001	*r* = −0.81 *p* < 0.0001	*r* = −0.68 *p* = 0.002	*r* = 0.09 *p* = 0.70
Try/BCAAs	-	*r* = 0.89 *p* < 0.0001	*r* = −0.97 *p* < 0.0001	*r* = 0.97 *p* < 0.0001	*r* = 0.69 *p* = 0.002	*r* = 0.88 *p* < 0.0001	*r* = 0.75 *p* < 0.0001	*r* = −0.26 *p* = 0.29
Prolactin (ng/mL)	*r* = 0.89 *p* < 0.0001	-	*r* = −0.91 *p* < 0.0001	*r* = 0.93 *p* < 0.0001	*r* = 0.86 *p* < 0.0001	*r* = 0.92 *p* < 0.0001	*r* = 0.87 *p* < 0.0001	*r* = −0.36 *p* = 0.014
Dopamine (pg/mL)	*r* = −0.97 *p* < 0.0001	*r* = −0.91 *p* < 0.0001	-	*r* = −0.99 *p* < 0.0001	*r* = −0.78 *p* = 0.0001	*r* = −0.93 *p* < 0.0001	*r* = −0.80 *p* < 0.0001	*r* = 0.27 *p* = 0.29

## Data Availability

The data presented in this study are available on request from the corresponding author.

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
