# Peer review of "Peripheral Modulators of the Central Fatigue Development and Their Relationship with Athletic Performance in Jumper Horses"

_animals, 2021, doi:10.3390/ani11030743_

Round 1
Reviewer 1 Report
Dear Authors, please change the marked sentences in abstract and conclusion!

Author Response
-We thank Reviewer for his/her suggestion. We carefully checked the attachment and we modified the sentence both in abstract and in conclusion accordingly.
Reviewer 2 Report
The approach of the study appears very original. The contents of the manuscript are quite interesting by his methodology and through the tools of quantification used. I find it interesting. I thus find that this paper definitively delivers results that will surely be of interest to the readership of the journal Animals.
However, some corrections are needed.
Introduction
L43 – It should be added which disciplines because not all of them need the high technical skills of the horse (ex. endurance).
L45 – when blood lactate is mentioned the word concentration should be added. (blood lactate concentration).
L48 – because performance monitoring is highly developed in race horses 2-3 sentences should be added about this. For example very novel techniques are used nowadays in race horses ex. changes in PBMCs proliferation and activity or cytokines mRNA expression.
L53 – the statement should be mitigate because there are a lot of papers connected with equine exercise physiology.
Materials and methods
L103 - was the level of fitness similar in all horses? What it was?
L116 – the change to The
L163 – I don’t understand why 3 tubes were collected for serum evaluation. In the future studies prolactin and dopamine should be measured from 1 tube to avoid high blood volume sampling.
Results
L194 - 196 – the sentence should be rewrite because is hard to follow
L199 – for better understanding the sentence should be rewrite to “…total protein and glucose concentration was unchanged after….”.
Figure 1 – of warm-up and competition
Figure 2 & 3– maybe use of asterixis and lines will be more reader-friendly
Discussion
L230 – the stress responses depend on type of exercise (ex. in endurance and race horses). It should be mentioned here.
L242 – it should be added that clearance of LAC takes 30 min in well-trained horses.
L248 – it is documented that the stress response after training sessions and competitions in horses is different. Thus, it should be mentioned here.
L 263-264 –the most developed techniques are nowadays used in race training such as infrared thermography which measures the changes in muscle temperature but also it correlates with lactate concentration.
Author Response
Comments and Suggestions for Authors
The approach of the study appears very original. The contents of the manuscript are quite interesting by his methodology and through the tools of quantification used. I find it interesting. I thus find that this paper definitively delivers results that will surely be of interest to the readership of the journal Animals.
However, some corrections are needed.
-We thank Reviewer for his/her positive comments and valuable suggestions. We improved the manuscript thanks to Reviewer’s suggestions.
Introduction
L43 – It should be added which disciplines because not all of them need the high technical skills of the horse (ex. endurance).
-We thank Reviewer for his/her valuable suggestion. We specified “Animals competing in equestrian disciplines, particularly in show jumping, are required to have high technical skills.”
L45 – when blood lactate is mentioned the word concentration should be added. (blood lactate concentration).
-We thank Reviewer for his/her suggestion. We change blood lactate with blood lactate concentration throughout the manuscript, as suggested.
L48 – because performance monitoring is highly developed in race horses 2-3 sentences should be added about this. For example very novel techniques are used nowadays in race horses ex. changes in PBMCs proliferation and activity or cytokines mRNA expression.
-We thank the Reviewer for his/her suggestions. We added more information on performance monitoring in race horses as suggested. In particular, We wrote “Nowadays, new markers of performance are used in race horses including the evaluation of peripheral blood mononuclear cells proliferation and activity or cytokines mRNA expression. Changes in immune cell proliferation, lymphocyte populations, and monocyte functionality have been described in trained and untrained racehorses after exercise confirming the creation of an anti-inflammatory environment in well-trained horses [2]. It has been suggested that long-distance endurance rides involve strenuous effort, which induces numerous changes in the horse’s body, including the exercise-induced acute-phase response [3]. Moreover, regular physical activity results in the decrease of pro-inflammatory states [3].”
L53 – the statement should be mitigate because there are a lot of papers connected with equine exercise physiology.
-We thank the Reviewer for his/her suggestion. We mitigated the sentence.
Materials and methods
L103 - was the level of fitness similar in all horses? What it was?
-We specified in material and methods section that “All horses enrolled in the current study had the same level of training, similar fitness, and the same experience of jumping competition.”
L116 – the change to The
-We thank Reviewer for pointing us out the mistake. We corrected it.
L163 – I don’t understand why 3 tubes were collected for serum evaluation. In the future studies prolactin and dopamine should be measured from 1 tube to avoid high blood volume sampling.
-We used 3 tubes for blood collection as kits used for the assessment of the parameters’ concentration required different centrifugation characteristics.
Results
L194 - 196 – the sentence should be rewrite because is hard to follow
-We thank reviewer for his/her suggestion. We rewrote the sentence as following “As showed in Figure 1, higher HR values were found at TPOST5 compared to T0 and TPOST30 at each step of competition (P<0.0001). Moreover, highest HR values were recorded during the course and at the outbound”
L199 – for better understanding the sentence should be rewrite to “…total protein and glucose concentration was unchanged after….”.
-We thank Reviewer for his/her suggestion. We changed the sentence as suggested.
Figure 1 – of warm-up and competition
-We thank Reviewer for his/her comment. We changed the Figure capture accordingly.
Figure 2 & 3– maybe use of asterixis and lines will be more reader-friendly
-We thank Reviewer for his/her comment. We changed the Figures accordingly.
Discussion
L230 – the stress responses depend on type of exercise (ex. in endurance and race horses). It should be mentioned here.
-We mentioned it as suggested.
L242 – it should be added that clearance of LAC takes 30 min in well-trained horses.
-We wrote “…it is known that the clearance of lactic acid from the muscles and bloodstream takes 30 min after exercise in well trained horses.”
L248 – it is documented that the stress response after training sessions and competitions in horses is different. Thus, it should be mentioned here.
-We mentioned that the stress response after training sessions and competitions in horses is different, as suggested.
L 263-264 –the most developed techniques are nowadays used in race training such as infrared thermography which measures the changes in muscle temperature but also it correlates with lactate concentration
-We thank Reviewer for his/her suggestion. We added this valuable information and related references (Witkowska-Piłaszewicz O, et al., Animals (Basel). 2020 Nov 9;10(11):2072).
Reviewer 3 Report
Review of the manuscript ID: animals-1131488, entitled „Peripheral modulators of the central fatigue development and their relationship with athletic performance in jumper horses”.
The main objective of this work was to evaluate relationship between the physical performance of competing jumping horses and some blood serotoninergic modulators and neurohormonal factors. Generally, the topic of the work is interesting. Achieving a good performance by the trained horse requires full adaptation of the organism to the specific type of exercise. It is still necessary to test the usefulness of new markers of horse fitness.
The work is written in understandable language.
Generally, Introduction and Materials and Methods are prepared correctly. However, in Statistical analysis is written that some indices of athletic performance in studied horses have been analysed (line 184). It is mentioned in Results, also (line 208) but nowhere else. No indices exist in Table 1 and they are not discussed. It should be completed because athletic performance is mentioned in the title of this manuscript. Moreover, Authors discussed dopamine level in peripheral blood plasma (lines 303-318) as a brain derived catecholamine only. In my opinion, the main source of dopamine in circulating blood could be the activity of adrenal medulla. Please, see: Podolak et al.: Comparison of the blood plasma catecholamines level in Thoroughbred and Arabian horses during the same-intensity exercise. Polish Journal of Veterinary Sciences 2006, 9 (1), 71-73.
Therefore, Minor revision is necessary.
Detailed comments:
in line 116, change “the next” to “The next”
Author Response
Comments and Suggestions for Authors
Review of the manuscript ID: animals-1131488, entitled „Peripheral modulators of the central fatigue development and their relationship with athletic performance in jumper horses”.
The main objective of this work was to evaluate relationship between the physical performance of competing jumping horses and some blood serotoninergic modulators and neurohormonal factors. Generally, the topic of the work is interesting. Achieving a good performance by the trained horse requires full adaptation of the organism to the specific type of exercise. It is still necessary to test the usefulness of new markers of horse fitness.
The work is written in understandable language.
-We thank Reviewer for his/her positive comments.
Generally, Introduction and Materials and Methods are prepared correctly. However, in Statistical analysis is written that some indices of athletic performance in studied horses have been analysed (line 184). It is mentioned in Results, also (line 208) but nowhere else. No indices exist in Table 1 and they are not discussed. It should be completed because athletic performance is mentioned in the title of this manuscript. Moreover, Authors discussed dopamine level in peripheral blood plasma (lines 303-318) as a brain derived catecholamine only. In my opinion, the main source of dopamine in circulating blood could be the activity of adrenal medulla. Please, see: Podolak et al.: Comparison of the blood plasma catecholamines level in Thoroughbred and Arabian horses during the same-intensity exercise. Polish Journal of Veterinary Sciences 2006, 9 (1), 71-73.
-We thank Reviewer for his/her positive comments and suggestions. The indices of performance considered in the current study are heart rate, HR; rectal temperature, RT; blood lactate concentration; blood glucose concentration. We assessed the correlation among the indices of athletic performance (e.g. HR, RT, blood lactate concentration, blood glucose concentration), the peripheral modulators of serotorinergic function (e.g. tryptophan; leucine; valine; Try/BCCAs ratio; NEFAs) and neurohumoral factors (e.g. domanine and prolactin)of in jumper horses throughout monitoring period.
We read and cited the paper by Podolak et al., Polish Journal of Veterinary Sciences 2006, 9 (1), 71-73, and we added the sentence “It has been suggested that the main source of dopamine in circulating blood could be the activity of adrenal medulla [45].” in the discussion section.
Therefore, Minor revision is necessary.
Detailed comments:
in line 116, change “the next” to “The next”
-We thank Reviewer for pointing us out the mistake. We corrected it.